

# Species interactions in an Andean bird–flowering plant network: phenology is more important than abundance or morphology

Oscar Gonzalez[1,2,4] and Bette A. Loiselle[1,3]

[1] Wildlife Ecology and Conservation, University of Florida, Gainesville, FL, United States of America
[2] Grupo Aves del Peru, Lima, Peru
[3] Center for Latin American Studies, University of Florida, Gainesville, FL, United States of America
[4] Department of Natural Sciences, Emmanuel College, Franklin Springs, GA, United States of America

Corresponding author
Oscar Gonzalez, pajarologo@ufl.edu

## ABSTRACT

Biological constraints and neutral processes have been proposed to explain the properties of plant–pollinator networks. Using interactions between nectarivorous birds (hummingbirds and flowerpiercers) and flowering plants in high elevation forests (i.e., "elfin" forests) of the Andes, we explore the importance of biological constraints and neutral processes (random interactions) to explain the observed species interactions and network metrics, such as connectance, specialization, nestedness and asymmetry. In cold environments of elfin forests, which are located at the top of the tropical montane forest zone, many plants are adapted for pollination by birds, making this an ideal system to study plant–pollinator networks. To build the network of interactions between birds and plants, we used direct field observations. We measured abundance of birds using mist-nets and flower abundance using transects, and phenology by scoring presence of birds and flowers over time. We compared the length of birds' bills to flower length to identify "forbidden interactions"—those interactions that could not result in legitimate floral visits based on mis-match in morphology. *Diglossa* flowerpiercers, which are characterized as "illegitimate" flower visitors, were relatively abundant. We found that the elfin forest network was nested with phenology being the factor that best explained interaction frequencies and nestedness, providing support for biological constraints hypothesis. We did not find morphological constraints to be important in explaining observed interaction frequencies and network metrics. Other network metrics (connectance, evenness and asymmetry), however, were better predicted by abundance (neutral process) models. Flowerpiercers, which cut holes and access flowers at their base and, consequently, facilitate nectar access for other hummingbirds, explain why morphological mis-matches were relatively unimportant in this system. Future work should focus on how changes in abundance and phenology, likely results of climate change and habitat fragmentation, and the role of nectar robbers impact ecological and evolutionary dynamics of plant–pollinator (or flower-visitor) interactions.

## INTRODUCTION

Interactions between flowering plants and their animal visitors are frequently focused on mutualistic encounters (*Bascompte & Jordano, 2014*). In these mutualisms, plants provide food resources (i.e., pollen, nectar), while animals provide pollinator services. Yet, these interactions are not always mutualistic. For example, animals may visit flowers and fail to effectively transfer pollen, as in the case of "nectar-robbers," which bypass reproductive parts of the flower via cutting a hole at the base to access nectar resources (*Rojas-Nossa, 2013*; *Maruyama et al., 2015*; *Rojas-Nossa, Sánchez & Navarro, 2016*). The presence of nectar-robbers in a system can change the dynamics and structure of plant–flower visitor networks. In these systems, the number and diversity of flower visitors to certain floral resources may increase as more visitors gain access to nectar, although the effectiveness of pollination may remain the same or even decline.

Plant–pollinator (or plant–flower visitor) networks have been relatively well-studied in recent years (*Lewinsohn et al., 2006*; *Burkle & Alarcon, 2011*; *Dalsgaard et al., 2011*; *Bascompte & Jordano, 2014*). These networks are almost always characterized by many fewer interactions than are possible and asymmetries (e.g., *Jordano, Bascompte & Olesen, 2003*; *Vazquez & Aizen, 2004*). Mutualistic networks such as plant–pollinator networks and plant–frugivore networks, often tend towards a nested structure. In the case of a bipartite network, nestedness is higher where more specialist species of one level interact with a few species in the other level, and this subset of species with few links are then shared with more generalist species (*Bascompte et al., 2003*; *Vázquez & Aizen, 2006*; *Bascompte, 2010*; *Thébault & Fontaine, 2010*). Mutualistic networks also have been found to be compartmentalized or modular with groups of species well connected to each other, but less connected to others in the network, usually when they have a large number of species (e.g., *Olesen et al., 2007*).

Recent research suggests that how networks are structured may influence their stability and co-evolutionary dynamics (*Bascompte & Jordano, 2007*; *Ebeling, Klein & Tscharntke, 2011*; *Allesina & Tang, 2012*; *Nuismer, Jordano & Bascompte, 2013*; *Suweis et al., 2013*; *James et al., 2015*). Thus, understanding which factors explain the observed interactions and structural properties of the network are key to predicting sensitivity of networks to perturbations, such as loss of species or changes in environmental conditions. Species extinction can be preceded by the extinction of species interactions, so this study contributes to show how network theory can help to explain the web of life in an ecosystem (*Bascompte & Jordano, 2014*). In recent years, new analytical approaches have facilitated asking questions about the processes that drive network properties (*Vazquez, Chacoff & Cagnolo, 2009*; *Encinas-Viso, Revilla & Etienne, 2012*; *Winfree et al., 2014*; *Vizentin-Bugoni, Maruyama & Sazima, 2014*; *Olito & Fox, 2015*). Two main hypotheses—neutrality and biological constraints—have emerged in these network studies. The neutrality hypothesis suggests that random interactions among species best explain network structure, such that relative species abundances predict interaction frequencies and can explain resultant structural properties (*Vazquez & Aizen, 2004*). In contrast, the biological constraints hypothesis suggests that interactions are shaped by species' traits or phenological patterns (*Jordano, Bascompte & Olesen, 2003*; *Vazquez, Chacoff & Cagnolo, 2009*; *Olesen*

*et al., 2011*). In the former, trait mismatches can result in "forbidden interactions" that impede or prohibit interactions among network members, such as when the length or width of the flower opening prohibits animal access to the nectar resources (*Jordano, Bascompte & Olesen, 2003*; *Olesen et al., 2011*). In the latter, phenological mismatches occur when animal presence in an area does not overlap the time when specific plants are flowering (*Vazquez, Chacoff & Cagnolo, 2009*).

Support for the neutral or biological constraints hypotheses have been mixed (e.g., see *Olito & Fox, 2015*). While information from species relative abundances (neutrality) and phenology (biological constraints) predicted components of network structure in plant–pollinator networks from Argentinean shrub land (connectance, nestedness, evenness and interaction asymmetry), neither of these hypotheses explained the observed frequencies of species interactions (*Vazquez, Chacoff & Cagnolo, 2009*). In contrast, in a hummingbird–flowering plant network, forbidden interactions from phenological or morphological mismatches were found to drive frequencies of observed interactions (*Maglianesi et al., 2014*; *Maruyama et al., 2014*; *Vizentin-Bugoni, Maruyama & Sazima, 2014*), although they were not important predictors of network structure (e.g., nestedness, connectance, specialization, evenness and asymmetry; see *Maglianesi et al., 2014*; *Vizentin-Bugoni, Maruyama & Sazima, 2014*). Similarly, phenological mismatches, in combination with relative abundances of network members, played a role in explaining interaction frequencies between nectarivorous sunbirds and flowering plants (*Janecek et al., 2012*).

Here, we extend these studies to investigate the drivers of species' interactions and network structure in a bird–flowering plant network in "elfin" forests located within the high Andes of Peru (*Brack & Mendiola, 2000*). Elfin forests, like other highland sites, are characterized by flowering plants adapted largely for bird pollination, as cold temperatures and often wet conditions limit insect abundance and activity (*Dalsgaard et al., 2009*; *Lloyd et al., 2012*). In mainland Americas these forests, while dominated by hummingbirds of various sizes and bill morphologies, also are frequented by *Diglossa* flowerpiercers (*Ramirez et al., 2007*). Flowerpiercers feed extensively on nectar, but may offer limited pollinator services as they frequently access flowers via holes they cut at the base of the corolla using their modified beaks (*Rojas-Nossa, 2013*). Their presence in the system may lessen the importance of morphological constraints in shaping interactions and structural properties of the network as they create opportunities for short-billed hummingbirds to also access flowers with long corollas. Thus, elfin forest networks may not fit the patterns reported earlier where interaction frequencies of networks are predicted by morphology of network members (*Maruyama et al., 2014*; *Vizentin-Bugoni, Maruyama & Sazima, 2014*; *Vizentin-Bugoni et al., 2016*).

By combining information on flower visits, flower phenology, bird (hummingbirds and flowerpiercers) and plant abundance, we address the following questions: (1) How are bird–flowering plant networks of elfin forest structured?, (2) Are observed interaction frequencies and network structural properties driven by biological constraints (morphological and/or phenological mismatches) or neutral processes (i.e., species relative abundance)?, and (3) How do visits by birds that offer little to no pollinator services affect network properties?

## METHODS

### Study area

Our study was conducted in the high elevation Andean forests of Peru known as "elfin forest" in Unchog, Huanuco Department, within the Carpish Mountains (9°42′32.33″S, 76°9′39.13″W; 3,700 m.a.s.l.). The elfin forest of Unchog is located within the transition between cloud forest and puna grassland. The area is characterized by a dry (May–September; <150 mm rain/month) and wet (October–March, >200 mm rain/month) season (Fig. S1) The study area is known to harbor a number of endemic bird (*Parker & O'Neill, 1976*; *Young et al., 2009*) and plant species (*Beltran & Salinas, 2010*).

Within the Unchog area, we sampled birds and plants in three elfin forest sites that had continuous vegetative cover and were ∼8 ha each—Unchog, Magma and Simeompampa; sites were from 0.6 to 1.7 km apart from each other and intervening habitat between these elfin forest patches was dominated by pasture and shrub land. To explore similarity between sites, we calculated pairwise Sorenson dissimilarity indices for plant species observed in the three sites (see *Trojelsgaard et al., 2015*) where values close to 0 indicate very similar community composition and values close to 1 indicate very distinct communities. For our sites, Sorenson values ranged from 0.13 to 0.25, indicating very similar plant composition. Further, the flower-visiting bird communities were also very similar. Therefore, sites were combined for network analysis due to the overlap in plant and bird species and the likely non-independence of the sites. This combination of sites increased power to characterize network with increased sample size.

This research was conducted under permits of the Peruvian government, Resolucion 151-2014-MINAGRI-DGFFS-DGEFFS and Resolucion 182-2012-AG- DGFFS-DGEFFS and the approval of the Institutional Animal Care & Use Committee of the University of Florida; IACUC Study #201105714.

### Behavioral observations

We quantified the flowering plant–bird network with direct observations on birds and plants using transects and focal plant watches (*Ortiz-Pulido et al., 2012*). These observations occurred between May 2011 and August 2014. Bird–plant interactions were observed using transects in the elfin forest patches approximately weekly during May–July 2011, February 2012, July–November 2012, January–July 2013, September 2013, and November 2013. During these visits, one of us (OG) walked along set transects inside the forest and along forest edges observing birds and recording which plants and how many flowering plants they visited during visits to the sites. If the bird visited more than one flower on a given plant during a visit, this was still scored as a single visit. In January 2014 and from May–August 2014 we recorded all visits to the flowering plants visited by birds using focal plant watches during 30-minute blocks. During these 30 min observation periods, multiple individual plants and plant species were simultaneously observed. Focal plant observations were centered on plant species exhibiting typical floral traits found in bird pollination syndrome (*Willmer, 2011*), and those that were known or suspected to be visited by birds based on previous observations (e.g., see *Maruyama et al., 2013*). We distributed these observations among the sampling areas in points at least 100 m apart. The time of observation for each

species was proportional to the abundance of the plant species. The combined observations of birds from transects and focal plants were used to build the interaction network (*Walther & Brieschke, 2001*).

We spent a total of 190 h (150 h in dry and 40 h in wet season) observing interactions over 79 days; 73 h, 52 h, and 65 h in Unchog, Magma, and Simeompampa, respectively. This effort was divided between transects (79%) and focal plant watches (21%); 50.7% and 49.3% of observed interactions were recorded by transects and focal plant watches, respectively. More time was spent during the dry season both because of increased flower abundance as well as logistics of working in the area.

## Plant phenology and abundance

We used transects to record abundance and phenology of flowers in the three sampling sites. We set up one transect per study site; these transects were sampled once a month at times when behavioral observations occurred. The presence or absence of flowers on a monthly basis was used to characterize phenology for each plant species. We counted the numbers of flowers per individual plant, or estimated the number of flowers by counting a subsample of flowers and then extrapolated to the whole plant for plants with >50 flowers. We converted the number of flowers to flowers per ha based on area sampled; in some transects, we corrected for effective area sampled given steep topography and inability to sample some areas at a 20 m width. We used flower density as a measure of plant relative abundance, as it has been shown to be a better estimator than the density of individual plants due to the high variability of flowers per plant (*Vazquez, Chacoff & Cagnolo, 2009*). For this network analysis, data were combined across sites and years due to similarities in species composition among sites and because sample sizes did not warrant more detailed examination of spatial and temporal patterns

## Nectarivorous bird phenology and abundance

To determine the phenology of birds at sites on a monthly basis, we scored presence or absence of birds based on point-counts, ad-hoc and behavioral observations, and mist-netting activities. To estimate overall relative abundance of bird species, we relied on mist-netting activities. We used 10–15 mist-nets (6 m or 12 m length, 36 mm mesh) by sampling bout in all the sites to capture birds, collect pollen when present from bills for further studies, and measure bill length and other morphological characters. Nets were distributed along forest edge and within the forest interior; nets were opened on one day per month overlapping periods where behavioral observations or focal plant watches occurred. Over the course of the study, mist-nets were opened a total of 2,399 mist-net hours (one 12-m net open 1 h = 1 mist-net hour). Vegetation height in the study area is 5 m on average and, thus, most birds that use the forest are expected to be captured using mist-nets. We recognize that not all birds are equally captured by mist-nets (e.g., *Remsen & Good, 1996*), and thus estimates may be biased. Nonetheless, in montane forest mist-netting is widely used as a recommended method for bird assessment (*Lloyd et al., 2012*; *Maglianesi et al., 2014*). As for plants, we combined the results among sites to characterize the bird community and bird–flower observations.

## Morphological measurements

We measured bill length and width (to nearest 0.1 mm) of birds that visited flowers using individuals captured in mist nets, supplemented by measurements from museums and published literature. We measured an average of 25 specimens per bird species. Since hummingbirds extend their tongues to access nectar inside the flowers, we added 1/3rd of the total length of the bird bill following *Vizentin-Bugoni, Maruyama & Sazima (2014)*; in a later paper *Vizentin-Bugoni et al. (2016)* recommend using a 80% tongue extension to correct bill measurement, although they found no difference in results when either 33% or 80% is used. We measured the length and the width (to nearest 0.1 mm) of flower corollas for plants visited by birds in the field, supplemented by measurements from herbarium specimens. The length was measured from the flower opening to the base of the nectar chamber, while the width was measured at the flower's widest aperture.

## Network description

Data on observed interactions at flowers were recorded as matrices with the bird flower visitors in columns, the plant species in rows and cell values representing the number of visits following *Jordano, Vázquez & Bascompte (2009)* and *Bascompte & Jordano (2014)*. We examined sampling completeness of nectarivorous birds and interactions in the study area using the Chao2 estimator in EstimateS version 9.1 (*Colwell, 2013*) following *Chacoff et al. (2012)* See Fig. S2.

We calculated the following network metrics: connectance, nestedness, interaction evenness and interaction asymmetry (*Bascompte & Jordano, 2014*). We also calculated a specialization index at the network level (H2') which is resilient to sample size and network size (*Blüthgen, Menzel & Blüthgen, 2006*). Connectance, which varies from 0 to 1, is the realized proportion of possible links in the network (i.e., if every bird visited flowers of every plant species, then connectance would equal 1). Nestedness provides a measure of the aggregation of the distribution of interactions in the network (*Nielsen & Bascompte, 2007*). To calculate nestedness, we used a weighted nestedness measure (WNODF) because WNODF has been found to be more robust in quantitative networks (*Almeida-Neto & Ulrich, 2011*). When WNODF is close to 0 there is no evidence of aggregation in the matrix, whereas as it approaches 100, the interactions are increasingly nested. Interaction evenness is based on Shannon's index following *Tylianakis, Tscharntke & Lewis (2007)* and provides a measure of the distribution of interactions in the network. High skewness in the distribution of interactions is indicative of an uneven network. Interaction asymmetry, which measures the strength and directionality of the interaction of one level to the other (birds and plants in this case), was calculated for plants and for birds separately; higher absolute values, from −1 to 1 indicate more uneven or skewed distribution of interaction frequencies. H2' measures specialization in the matrix based on the H index of Shannon–Wiener (*Blüthgen, Menzel & Blüthgen, 2006*). H2' describes how much the observed distribution of species interactions deviate from the frequency of the expected distribution. It ranges from 0 to 1; when H2' is closer to 1, the interactions are considered to reflect a high degree of specialization. Connectance (conn), nestedness (WNODF), evenness (interaction evenness), specialization (H2'), and interaction asymmetry (intrasymm) were

calculated using bipartite package version 2.05 in R (*Dormann, Gruber & Fründ, 2008*). See Table S1 for R source code.

## Interaction probability matrices

We built interaction probability matrices using the framework proposed by *Vazquez, Chacoff & Cagnolo (2009)* where interaction frequencies were assumed to be determined by relative abundances, temporal (phenological) overlap, and morphological overlap. As above, these probability matrices are based on the data compiled across the three study sites. Relative abundance probability matrices will provide a test of the neutrality hypothesis, while the latter two (phenology, morphology) provide a test of the biological constraints hypothesis in explaining observed network structure and interaction frequencies.

To develop a phenological interaction probability matrix (PhenMat), we first compiled matrices of temporal overlap for plants and birds. In these matrices, plant or bird species were rows and sampling months were columns with ones and zeros for presence and absence; the total number of months with simultaneous data on both plants and birds were 15. We then used matrix multiplication to obtain temporal overlap between birds and plants. This matrix of temporal overlap was normalized such that the matrix cells added up to a total of one; individual cells with higher values indicated greater temporal overlap, or probability of interaction, of any particular bird-plant pair.

An abundance interaction probability matrix (AbMat) was made in a similar way as the phenological matrix, compiling matrices of abundance for plants and for birds and overlapping them in the same months. Here the cells of the plant matrix were the number of individual flowers per ha, by species summed across the sites. The cells of the bird matrix were the number of individuals captured in mist-nets per 100 net-hours (*Maglianesi et al., 2014*). We multiplied the two abundance matrices and the product was normalized as explained above.

The morphology interaction probability matrix (MorMat) was generated to account for morphological mismatches in length of a bird's bill (as corrected to account for tongue, see above) and corolla length (*Maruyama et al., 2014*; *Vizentin-Bugoni, Maruyama & Sazima, 2014*). However, instead of using mean length, we used the probability of size overlap between ranges of flower length and bill length. We believe this approach is more realistic than a simple yes or no criterion because of existing intra-specific variation in morphological traits among individuals (*Gonzalez-Castro et al., 2015*; *Gonzalez-Varo & Traveset, 2016*). We first noted the range of a flower's length and the range of a bird's bill. If the lower limit of the bill's range was longer than the lower limit of the flower's length, the interaction was scored as 100% possible with a cell value of 1. If the upper limit of the bill's range was shorter than the lower limit of the flower's length, the interaction was considered impossible and a cell value of 0 was assigned. When there was overlap of ranges between the length of a bird's bill and the length of a corolla, we calculated the proportion of overlap and assigned that value to the cell. Furthermore, we considered some exceptions when the flower's width was expansive enough for a bird's head to enter the corolla. For example, the flowers of *Puya* are longer and wider than any of the bird's bills, so we considered that all bird species could visit *Puya* and assigned a value of 1 for

all possible interations with this flowering species. In cases where size overlap was zero, but the observed interaction frequency was not zero, we assigned an arbitrary value of $1 \times 10^{-8}$ which is less probable than any other case in the phenology and abundance matrices (*Gonzalez-Castro et al., 2015*). Further, the placement of zero in the probability matrix when the observed interaction value is not zero results in a failed calculation of the multinomial function (see next topic). As for other interaction probability matrices, we normalized this matrix so that cell values sum to 1.

We also considered the possibility that factors might act together to influence the observed bird-flower network To do this, we used matrix multiplication to create new interaction probability matrices for all possible combinations—AbMat*PhenMat, AbMat*MorMat, PhenMat*MorMat and AbMat*PhenMat*MorMat—and then normalized these new matricies so that the cells summed to one. Following *Vazquez, Chacoff & Cagnolo (2009)* we also included a "Null" probability matrix where all pairwise interactions in the matrix made of observed plant and animal species had the same probability of occurrence (i.e., all cell values in the matrix are equal to $1/IJ$, where $I$ and $J$ are number of plant and bird species in the network).

## Testing neutrality and biological constraints hypotheses

To test whether neutral processes or biological constraints best predicted observed interaction frequencies, we used a likelihood approach. Support for either of these hypotheses arises when the probability matrix can predict the observed interactions, such that higher probabilities of cells should match with higher frequencies of observed interactions (see *Vazquez, Chacoff & Cagnolo, 2009*; *Vizentin-Bugoni, Maruyama & Sazima, 2014*). Akaike information criteria (AIC) was used to compare the relative ability of these various hypothesized models to explain observed interactions. We assumed that probability of interaction between a given bird and flowering plant followed a multinomial distribution (*Vazquez, Chacoff & Cagnolo, 2009*). We used the number of species (44 in total), to determine the number of parameters used to weight different models' complexities when calculating AIC. So 44 was used when one factor was calculated (i.e., abundance), 88 if two factors (i.e., phenology and morphology) and 132 if three factors (i.e., phenology, morphology and abundance). As in *Vizentin-Bugoni, Maruyama & Sazima (2014)* we compared these results to those based on using the number of factors (abundance, phenology, morphology; either. e., 1, 2 or 3) to weight model complexity, and checked for differences. The function *dmultinomin* in the *stats* package R was used to calculate likelihood (*R Core Team, 2014*).

To determine the degree to which the hypotheses predicted network metrics, such as connectance, nestedness, or asymmetry, we used a randomization algorithm *mgen* from *bipartite* package in R (*Vazquez, Chacoff & Cagnolo, 2009*). Using the number of interactions actually observed, the randomization algorithm assigned interactions to each probability interaction matrix, including all combined interaction probability matrices, such that all species received at least one interaction (see *Vazquez, Chacoff & Cagnolo, 2009*). From these randomized matrices, we calculated network statistics (mean and 95% confidence intervals from 1,000 repetitions using function *confint* in *bipartite* package in R)

and compared the overlap with network statistics generated from our observed interaction matrix. If the observed metric values were found to be within the 95% confidence intervals of those generated from interaction probability matrices, we interpreted this to mean that factors (e.g., relative abundance, phenology, morphology, or their combination) could explain or were consistent with hypothesized explanations of drivers of mutualistic interactions at the community level.

### Effect of nectar-robbers on network properties

To examine the influence of nectar robbers on network properties, we recalculated all of the above matrices after removing interactions that likely did not result in pollination, such as visits to base of flowers through holes cut by *Diglossa* flowerpiercers or bees. This new interaction matrix is more equivalent to a pollinator–plant network than our bird–flowering plant network which included all flowering plant visits (*Maruyama et al., 2015*). This pollinator–plant network was then used to evaluate our third question that examined the impact of flowerpiercers on network structure and network properties as described above

## RESULTS

### General results

#### Bird-flower network

We observed a total of 17 bird species visiting flowers from 27 plant species in all our elfin forest sites combined. These observations included 278 pairwise interactions representing 74 unique interactions of bird visits to plants (Table S2). Avian flower visitors included 12 species of hummingbirds (Trochilidae), four species of flowerpiercers (*Diglossa*: Thraupidae) and one conebill (*Conirostrum*: Thraupidae). Flowering plants observed to be visited by birds included plants from 24 genera, 16 families and 14 orders (Fig. 1, and Table S3).

Mutualistic networks are typically characterized by many fewer observed interactions than possible (e.g., *Chacoff et al., 2012*), and this was also true here. We detected only 55.2% of the estimated interactions for the whole network using Chao2 (Fig. S2). Despite this, the observed number of unique interactions appeared to be reaching an asymptote with our sampling effort.

#### Interactions in the network

We found that *Metallura theresiae, Pterophanes cyanopterus* and *Diglossa mystacalis*, birds considered to be indicators of the elfin forest (*Parker, Stotz & Fitzpatrick, 1996*), were the most important bird species in terms of flowering plant interactions (Fig. 1). *Metallura* visited a total of 26 species, while *D. mystacalis* visited 10 and *Pterophanes* visited 4 species, respectively. Among plants, *Brachyotum lutescens, Tristerix longebracteatus* and *Fuchsia decussata* were the species with the most interactions with visits from 8, 7, and 7 bird species, respectively (Fig. 1). When compared to the "null" model, the bird–flowering plant network was found to be significantly less connected and more nested (Figs. 2A and 2B). In addition, the network was significantly less even, more specialized and exhibited greater asymmetry among bird or plant species than expected (Figs. 2C–2F).

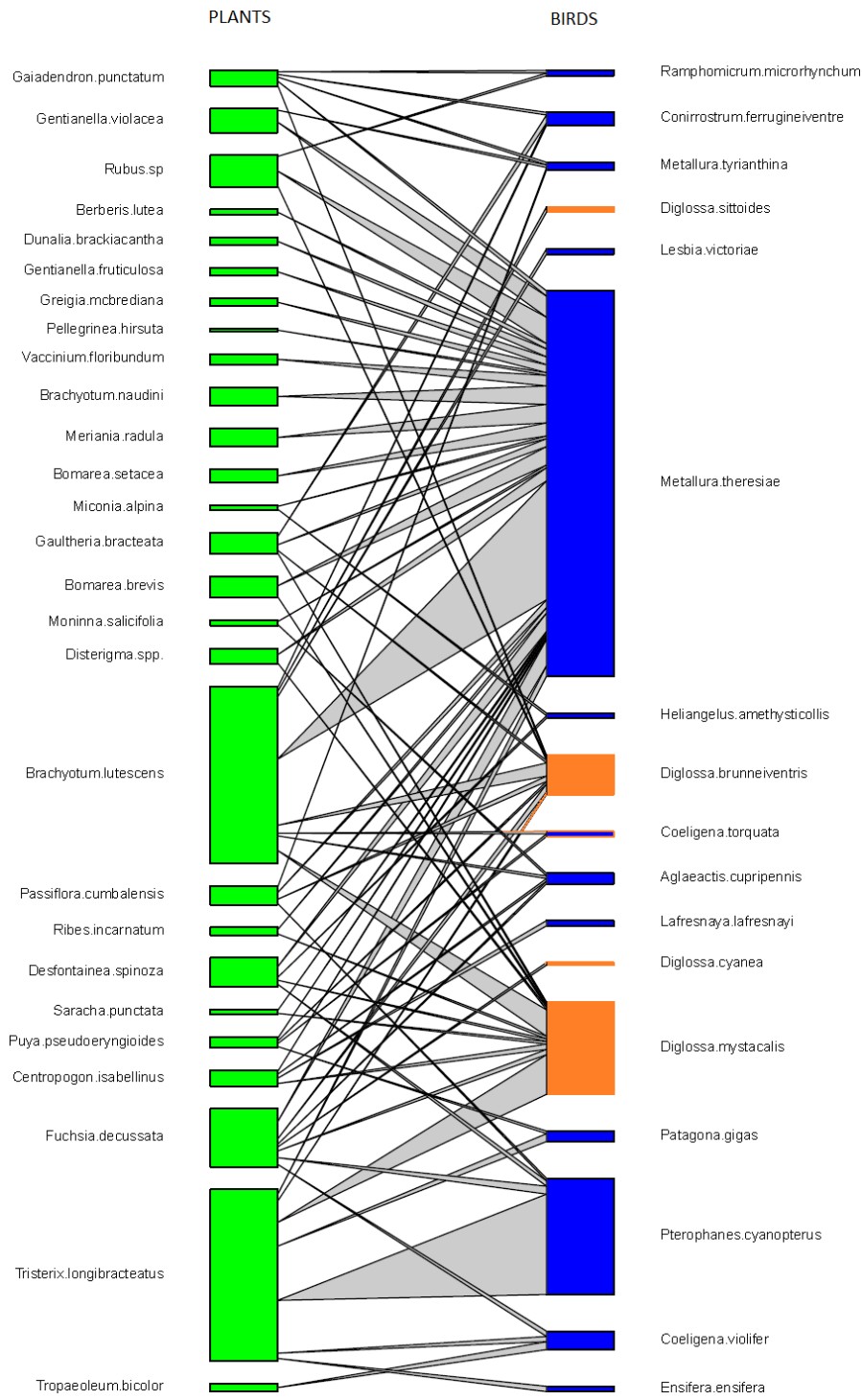

**Figure 1  Pairwise interactions in the bird–flowering plant visitation network in the elfin forest of Unchog (Peruvian Andes).** Each green box represents a plant species; blue boxes are hummingbirds, orange are flowerpiercers. The lines represent the interactions and the thickness of the line reflects the number of interactions.

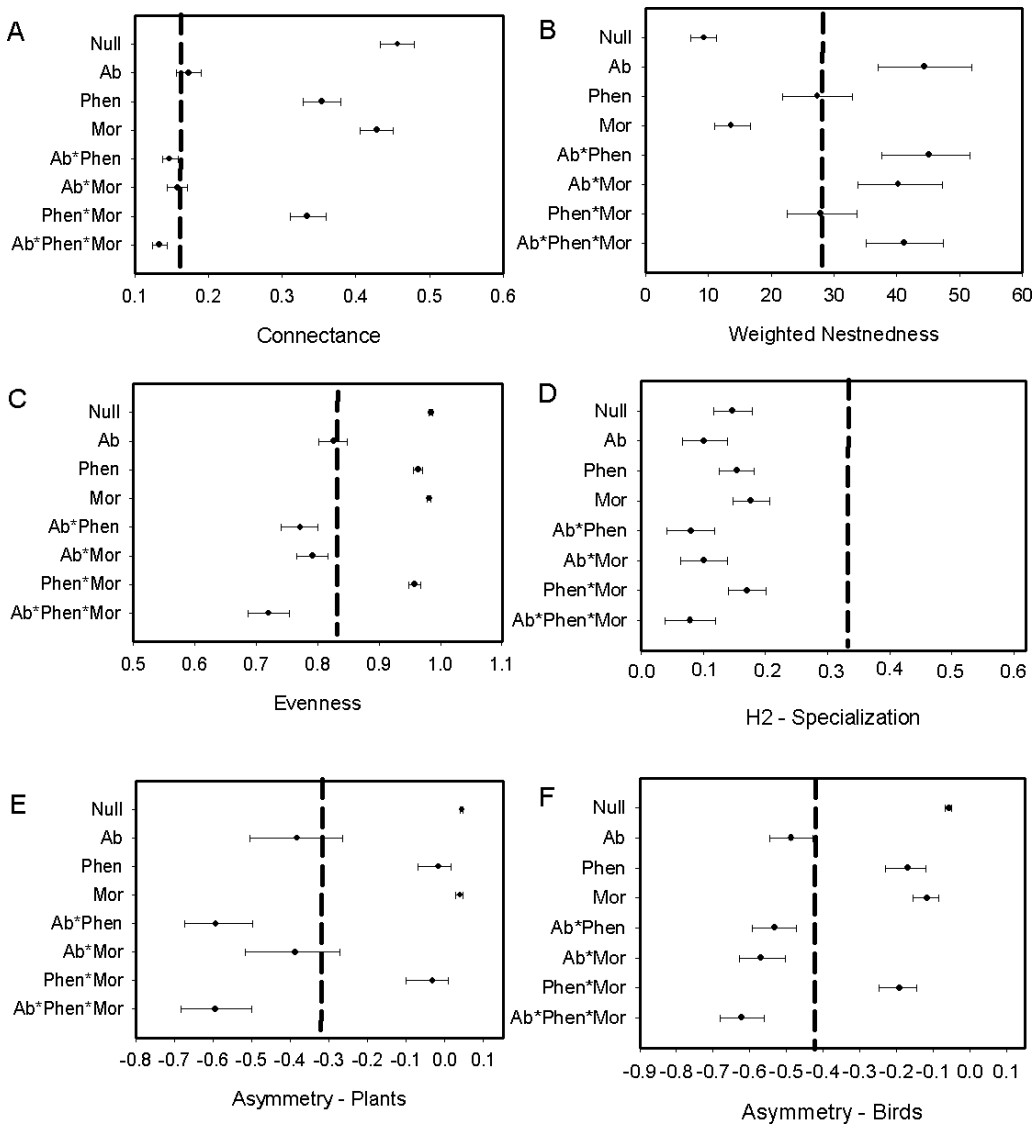

**Figure 2 Comparison of the network metrics produced by probability matrices (mean and 95% confidence intervals) and the observed network values for the bird–flowering plant network of the elfin forest.** The dashed vertical lines in each graph represents the value for the observed matrix. Matrix codes: Ab, Abundance; Phen, Phenology; Mor, Morphology; Null, Null matrix.

## Plant-visitation network determinants

We found that biological constraints as exemplified by phenology best explained the observed interaction frequencies using the likelihood approach (Table 1). This result suggests that the degree of temporal overlap among interacting players in a bird–flowering plant network is an important driver of the observed interaction frequencies. Results were consistent whether either the number of species or the number of matrices used as parameters in likelihood analysis to compare among models (Fig. S3).

Biological constraints, either through phenological constraints or phenology in combination with morphology, were found to explain nestedness in the elfin forest

**Table 1** **Difference of the AIC values between various models predicting observed interaction frequencies and the model with lowest AIC value.** Ab, Abundance; Phen, Phenology; Mor, Morphology; Null, Null matrix. The visitation network includes all bird species and the pollination network excludes interactions where birds did not visit flowers legitimately. In both matrices, phenology was the best predictor of interaction frequencies.

| Matrix | ΔAIC visitation network | ΔAIC pollination network |
|---|---|---|
| Phen | 0 | 0 |
| Ab | 428.2 | 180.3 |
| Null | 588.6 | 433.4 |
| Ab*Phen | 660.1 | 387.0 |
| Phen*Mor | 1262.9 | 122.9 |
| Ab*Mor | 1846.8 | 365.0 |
| Mor | 1912.1 | 575.9 |
| Ab*Phen*Mor | 2067.4 | 563.7 |

bird-plant network (Fig. 2B). Network structure also was found to be explained by neutral processes, as measured by relative abundance of birds and flowering plants, in some cases. For example, connectance within the network was predicted via a combination of relative abundance of interacting players and biological constraints (Fig. 2A). Further, relative abundance was found to explain evenness and relative abundance alone, or in combination with morphology, explained asymmetry (Figs. 2E and 2F). In contrast, neither biological constraints nor neutral processes were able to explain specialization (Fig. 2D).

## Difference in network properties with and without nectar-robbers

To investigate the influence of nectar robbers in the plant–bird network, we removed all interactions that involved observed 'illegitimate'' visits to flowers (i.e., birds entered flower at the base rather than through the corolla opening). This reduced the original network of 17 birds and 27 plants to 12 birds and 26 plants; all *Diglossa* species dropped out of the network given that all observations were from *Diglossa* or *Conirostrum* removing nectar from the base of flowers. Other interactions deleted were *Heliangelis amestyticollis–Desfontainia spinosa, Pterophanes cyanopterus–Passiflora cumbalensis, Metallura tyrianthina–Passiflora cumbalensis* and *Metallura theresiae* with *Desfontainia spinosa, Fuchsia decussata* and *Passiflora cumbalensis*. Removal resulted in the loss of all interactions with *Passiflora cumbalensis*, given that all observations to this flower occurred via the floral base and were not considered "legitimate".

The exclusion of illegitimate visits by primary and secondary nectar-robber birds resulted in a decrease of connectance, nestedness and evenness, but increase in specialization when comparing metrics with the null matrix and the original matrix. Asymmetry shifted in different ways with an increase for plants and decrease for birds (compare Figs. 2 and 3). Abundance and its combination with morphology were useful to explain evenness and asymmetry for plant and birds, while phenology in combination with morphology predicted connectance. However the exclusion of the nectar-robbers in the network did not change the influence of phenology as the "best" predictor of species interactions (Table 1).

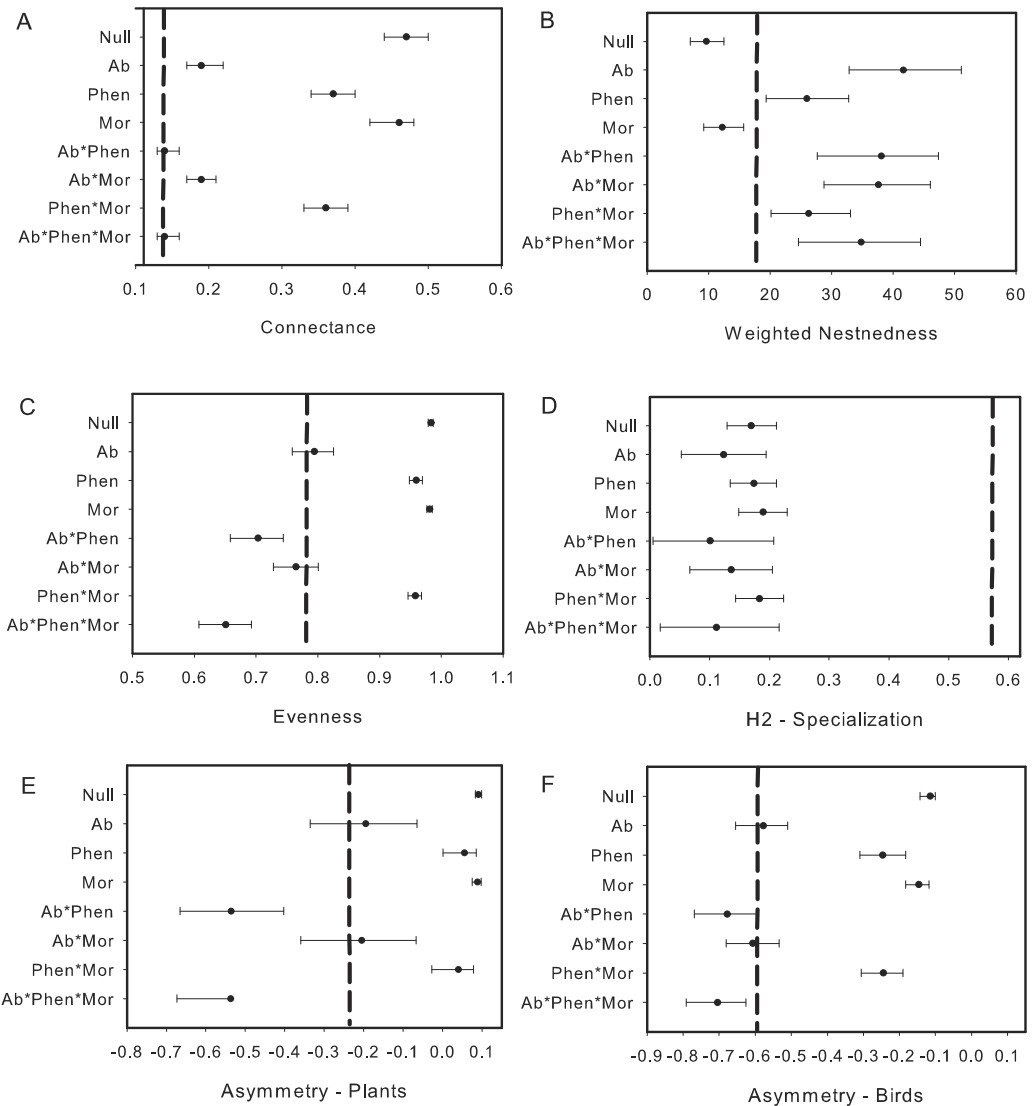

**Figure 3** Comparison of the network metrics produced by probability matrices after removal of interactions involving nectar-robbers. The dashed vertical lines in each graph represents the value for the observed matrix. Matrix codes: Ab, Abundance; Phen, Phenology; Mor, Morphology; Null, Null matrix.

## DISCUSSION

### Network structure

We found that bird–flowering plant networks in the elfin forests studied here are characterized by fewer interactions than those possible. These results are consistent with most other pollination networks studied (*Bascompte & Jordano, 2014*). As a coincidence, *Chacoff et al. (2012)* also observed about 55% of all possible interactions in a desert plant–pollinator network sampled over 4 years. Yet, despite their time investment, *Chacoff et al. (2012)* estimate that a five-fold increase in sample effort would be needed to even possibly detect 90% of the possible interactions. The sampling effort invested in our study (190 h) spread over multiple years matches or exceeds many other studies in bird–flowering

plant networks (e.g., *Rodriguez-Flores, Stiles & Arizmendi, 2012*; *Ortiz-Pulido et al., 2012*; *Maruyama et al., 2014*), but is considerably less than *Traveset et al. (2015)* and studies that use video-cameras to record interactions (*Maglianesi et al., 2014*; *Vizentin-Bugoni, Maruyama & Sazima, 2014*; *Vizentin-Bugoni et al., 2016*). The use of remote technology in flowering plant observational studies hold much promise, but are nonetheless, difficult or expensive to use in sites where flowering plant diversity is high.

Like several mutualistic networks, the elfin forest network was also found to be nested (Fig. 3B; see also *Rodriguez-Flores, Stiles & Arizmendi, 2012*; *Bascompte et al., 2003*). Our results, however, contrast with some studies in tropical dry forests (*Arizmendi & Ornelas, 1990*) and Atlantic forests (*Vizentin-Bugoni, Maruyama & Sazima, 2014*) where the plant–bird network was not nested using the same qualitative metric (WNODF). However in a more recent analysis an updated version of this Atlantic forest network was nested using a binary measure (*Vizentin-Bugoni et al., 2016*) *Bascompte et al. (2003)* suggest that increases in the number of species in networks might result in greater nestedness. *Sebastian-Gonzalez et al. (2015)* confirmed this increase in nestedness with greater number of species, but other studies found no such effect (*Cuartas-Hernandez & Medel, 2015*) or even reduced nestedness with increased number of species (*Ramos-Robles, Andresen & Diaz-Castelazo, 2016*). Indeed, when we reduced the network to only include species and observations that resulted in "legitimate" flower visits, we found nestedness values were lower although the network was still more nested than expected. In the elfin forest the abundance of the flowerpiercers facilitating access to hummingbirds would take out several "forbidden links" as limitations, diminishing nestedness. Sampling effort did not likely bias our estimate of nestedness given that WNODF is known to be a robust estimator for nestedness in nectarivore bird-plant networks (*Vizentin-Bugoni et al., 2016*, but see *Costa et al., 2016*).

The important species in networks, based on their abundance and frequency of interactions, often provide insights about the ecological or evolutionary implications of the network (*Bascompte & Jordano, 2007*). In this system the most abundant flower visitor in elfin forests (*Metallura theresiae*) also had the greatest number of connections and interacted with the most flowering plant species (Fig. 1). *Metallura theresiae* is quite aggressive and its behavior may interfere with other flower visitors, and thus, may affect visitation rates (*Justino, Maruyama & Oliveira, 2012*). In contrast, the plant which had the greatest number of flowers in this system, *Brachyotum lutescens*, did not have the greatest number of flower visits nor the greatest number of visiting species. Similar results were reported by *Rodriguez-Flores, Stiles & Arizmendi (2012)* in a plant–hummingbird network in Colombia, where hermit hummingbirds were the most abundant birds, visiting the greatest number of plant species in the lowland Amazon.

In the elfin forest system we found that abundance models combined with phenology or morphology can explain network connectance, as well as evenness and asymmetry for both plants and birds. Nestedness was predicted by phenology and phenology with morphology, matching results from an insect-plant network in a subalpine community (*Olito & Fox, 2015*). In contrast, in plant–hummingbird networks in the Brazilian Atlantic forest (*Vizentin-Bugoni, Maruyama & Sazima, 2014*) neither abundance, morphology nor

phenology were associated with network metrics, except the mixed model of phenology, morphology and abundance for the asymmetry of the birds in the network.

## Observed interaction frequencies

In this study, the observed interactions, either with the full suite of avian flower visitors or the reduced set of "legitimate" visitors, were best explained by phenology. Greater phenological overlap in birds and plants led to greater number of interactions between pairs of species. The importance of phenology in explaining pairwise interactions has also been found in other studies, but was still found to be a poor predictor of observed interactions in some cases (*Encinas-Viso, Revilla & Etienne, 2012*; *Olito & Fox, 2015*).

In contrast, morphology alone, or morphology interacting with phenology, have been found to explain observed pairwise interactions in some hummingbird-plant networks (*Maglianesi et al., 2014*; *Maruyama et al., 2014*; *Vizentin-Bugoni, Maruyama & Sazima, 2014*). The importance of phenology in driving interactions highlight the potential vulnerabilities of these mutualism networks to climate change, which can alter phenological patterns (*Dalsgaard et al., 2011*; *Rafferty, CaraDonna & Bronstein, 2015*). Elfin forests which are located at the top of tropical mountains are likely to be particularly impacted by climate change and, thus, as phenological patterns change, nectarivorous birds, a number of which are endemic, may face lowered availability of resources and potential invasion of competitors (*Sekercioglu, Primack & Wormworth, 2012*).

## The nectar robber effect in the network

Morphological constraints were not an important driver in our system, or only were important when combined with abundance or phenology for some network metrics. This result is likely due to the presence of *Diglossa* flowerpiercers. The opportunities for morphological constraints to operate in this system are many as several flowering plant species have corollas that exceed the length of a number of flower visitors. Yet, the forbidden interactions in this network, which hypothetically should restrict access to nectar for small-billed birds for a number of flowering species, were allowed. *Diglossa*, which cut holes in base of flowers to gain access to nectar, act as facilitators for other species (hummingbirds with small bills) that would not be able to access to long-corolla flowers (*Bruno, Stachowicz & Bertness, 2003*); large bees are also known to cut holes at base of flowers in this system and may also serve as "facilitators". Consequently, connectance in the network increases with flowerpiercers in the system (Fig. 2A with Fig. 3A). In contrast, we found that network specialization increased markedly when nectar-robbers were removed from the network (Fig. 2D with Fig. 3D).

In this study several factors might influence our results. First, we observed only about 55% of all possible pairwise interactions. If our system were undersampled, including potentially "missing" interactions due to not capturing flowering events because of limited sampling, we might have been less likely to see biologically constrained interactions, and thus, may have overestimated their effect. However, we did find that the number of observed pairwise interactions appeared to be reaching an asymptote, suggesting we had sufficiently characterized the network. Second, as most observations were based on transects, we might

expect that abundance may emerge as a driver of network interactions and properties as abundant species may be sampled more often in focal plant watches. Nevertheless, the amount of interactions detected by transects was almost the same as the interactions detected by focal watch. The importance of relative abundance as a driver, however, did not play a large role in explaining observed interactions when compared to other factors. Third, we also did not examine the importance of body size of birds and nectar production in explaining network structure. Nectar production can be highly variable both within and among plants and is difficult to adequately measure when dealing with many plant species. Large-bodied birds, in particular, might focus more on plant species that produce more nectar. These factors might be especially important in explaining network properties such as specialization. Further studies would benefit from including additional predictors of networks.

In summary, in elfin forests biological processes were important in predicting observed interactions between flowering plants and birds, while neutral and biological processes interacted to explain network components. In particular, the importance of neutral processes (i.e., abundance) was the single best predictor for four of six network metrics in networks with and without illegitimate interactions. However, the importance of phenology for both species interactions and network structure suggests that the ecological and evolutionary dynamics of networks are likely to be altered with climate change. As such, future studies should focus on how phenological changes, as well as changes in abundance impact network dynamics.

## ACKNOWLEDGEMENTS

We thank Camilo Diaz, Bernie Britto and Juan Diego Shoobridge for plant identification. OG thanks Jessica Burnett and J. Vizentin-Bugoni for their help in R codes.

### Funding

OG received funding for PhD studies from Fondo para la Innovacion, Ciencia y Tecnologia in Peru and World Wildlife Fund. Funding for fieldwork came from the Premio Nacional para la Investigacion Ambiental of the Ministerio del Ambiente of Peru, Optics for the Tropics, Royal Society for Bird Protection, Idea Wild and University of Florida's Tropical Conservation and Development Program field research fund. The funders had no role in study design, data collection and analysis, decision to publish, or preparation of the manuscript.

### Grant Disclosures

The following grant information was disclosed by the authors:
Fondo para la Innovacion, Ciencia y Tecnologia in Peru and World Wildlife.
Premio Nacional para la Investigacion Ambiental of the Ministerio del Ambiente of Peru, Optics for the Tropics, Royal Society for Bird Protection, Idea Wild.
University of Florida's Tropical Conservation and Development Program field research.

## Competing Interests

The authors declare there are no competing interests.

## Author Contributions

- Oscar Gonzalez conceived and designed the study, collected the field data, analyzed the data, prepared figures and tables, and wrote and reviewed drafts of the paper.
- Bette A. Loiselle conceived and designed the study, analyzed the data, prepared figures and contributed to writing and reviewing drafts of the paper.

## Animal Ethics

The following information was supplied relating to ethical approvals (i.e., approving body and any reference numbers):

Institutional Animal Care & Use Committee of the University of Florida; IACUC Study #201105714.

## Field Study Permissions

The following information was supplied relating to field study approvals (i.e., approving body and any reference numbers):

Peruvian government, Resolucion 151-2014-MINAGRI-DGFFS-DGEFFS and Resolucion 182-2012-AG- DGFFS-DGEFFS.

## Data Availability

Figshare: https://figshare.com/s/3903e803bedc61aeee81.

## Supplemental Information

Supplemental information for this article can be found online at http://dx.doi.org/10.7717/peerj.2789#supplemental-information.

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
