# Peer review of "Species interactions in an Andean bird–flowering plant network: phenology is more important than abundance or morphology"

_PeerJ, doi:10.7717/peerj.2789_

## Round 0.1 · original submission · Major Revisions

Your ms has been reviewed by two referees who agree this is an interesting study that deserves publication, but at the same time indicate a number of important concerns. I am sure all their valuable comments will be very useful to you when preparing the next version of the ms. Please consider all of them and justify in the next cover letter if you do not agree with some of the comments/suggestions.

Reviewer 1 ·

Basic reporting

overall a well conducted study, but I have a number of questions to the experimental design (see below), some questions / concerns regarding the biology and a few references that I think would make sense to cite in the present study.

L. 48-50. See also Maruyama et al. 2015. Nectar robbery by a hermit hummingbird: association to floral phenotype and its influence on flowers and network structure. Oecologia 178: 783-793.

L. 108: To make this more general and not just applicable to the Andes, see also Dalsgaard et al. 2009. Oecologia 159, 757-766.

L. 118. See also Vizentin-Bugoni et al. 2016. Journal of Animal Ecology 85, 262-272
See also “general comments” for more citations.


See more in "general comments". Hope you find the comments useful - and I hope to see the published in the near future.

Experimental design

L. 139-142. You sample three localities up to 1.7 km apart and pool them into one network. There is a matrix of pasture in between the sites. I would expect this to be able to structure the network, eg tendency that the sites form separate modules (even if having some species in common). Did you test if ok to pool these three networks together into one big network?
L. 148-151. You made observations based on both focal observations and walking along set transects inside the forest and along forest edges observing birds and recording which plants and how many flowers they visited during visits to the sites. If the bird visited more than one flower on a given plant during a visit, this was still scored as a single visit. Based on this procedure I would expect abundance to be important structure the resulting network, as abundant species will be observed more often just by chance. In other system, eg the cited Vizentin-Bugoni et al. 2016 paper, purely used focal observations rather than transects. I would think this difference in sampling would cause the present study to bias toward abundance being important, which indeed was the identified results. Could you also elaborate / convince me that I am wrong that transect observations should cause abundance to become important in structuring the network? It is unsure how many observations were based on transects and how many based on focal observations. Could you please specify?
L. 185-188. Abundance of birds was estimated using mist-nets. Please make a note that these data may be biased, as you estimate the abundance of those birds flying the high of the mistnets. You recognize this yourself (L. 194-199). I would think the preferred way would be to estimate bird abundance using point counts and distance sampling. You seems to have collected these data too, but do not use them. Please explain why (could just be in the coverletter).
L. 205-207. Please see new reference from same authours now using a 80% tongue extension estimate (Vizentin-Bugoni et al. 2016. Journal of Animal Ecology 85, 262-272). This could cause a big difference.

Validity of the findings

see comments above to the "experimental design".

Additional comments

Abstract: please in the abstract define what you mean with “neutral processes”. That is not clear from reading the abstract only.
L. 59: “species interact with a subset of species” seems unfinished.
L. 63: “not to” should be “less to”.
L. 65-70: found the language too strong. Suggest using eg “may” rather than “has been shown to”.
L. 95: Suggest “hummingbird” instead of “bird”.
L. 108. Suggest adding “In mainland americas”, so become “In mainland Americas, these forest, while”.
L. 113-115: also because the flower pierces themselves are less restricted morphologically, right? But up until now I am a little unsure whether you include the flower pierces in your study, or you hypothesize that hummingbird-plant networks will be less structured by morphology because of flowerpierces being present. Important to make this 100% clear already here in the Introduction.
L. 119-124. I am unsure whether you also include the flower pierces in these questions, or whether it is only the hummingbirds and their plants.

L. 406: Sounds as if this is characteristics for all elfin forest in the world, but I guess this applies to your specific study, right? Make that clear please, eg “We found that…”.
L. 422: The study by Vizentin-Bugoni et al is from Atlantic rainforest, not savannah. In addition, please look at their 2016 study where the find that binary nestedness is significant but weighted nestedness is (as you cite) not nested.
L. 437: Please add “may” to ie “may affect”.
L. 448-451. Again, the studies by Vizentin-Bugoni et al (2014 , 2016) are not from the savannah. But actually there is a Brazilian savannah study on similar subject: Maruyama et al. 2014. Biotropica 46, 740–747. Be sure to which system you refer – the atlantic rainforest or the savannah. Both systems are welworth referring to as very similar studies to this one here.
L. 464. Climate change may also affect network structure, see the already cited paper by Dalsgaard et al. 2011. Plos One, also on hummingbird-plant networks.
L. 470-489. See also Maruyama et al. 2015. Nectar robbery by a hermit hummingbird: association to floral phenotype and its influence on flowers and network structure. Oecologia 178: 783-793.

In the Discussion, please discuss that you did not include the role of bird body size and nectar production, which are known to be correlated, ie determine interactions between hummingbirds and plants (eg Dalsgaard et al. 2009. Oecologia 159, 757-766). Do you think including this would change your results, and why or why not. Would like to see that in the discussion.

·

Basic reporting

This study explores a central question in community ecology: whether the interaction patterns observed on species interaction networks, are better explained by random species encounters (i.e. neutral theory) or by biological constraints (i.e. morphological or phenological mismathes). The key findings are well supported and the manuscript represents a relevant contribution for understanding the drivers of ecological network structure and is likely to be well received by the readership of PeerJ.
The data collection suffers from low standardization (through space and time) and sub-optimal sampling effort, nevertheless, I believe that the results are well supported by the data and further sampling effort would likely amplify the effect of the current findings.
The whole text is extremely clear and the relevant background literature is well articulated.
Overall the figures are informative and relevant but I have a few suggestions, namely:
Figure 1 could be considerably more informative if the two trophic levels were labelled, and if legitimate and illegitimate (flowerpiercers) visitors where marked (for example with different colors). The caption can also be improved by including information relative to the location of the study and a very brief explanation of what the boxes and lines represent.
I believe that figures 2 and 3 would be easier to interpret if they were merged into a single figure and if the same scale was used for panels reporting the same metric.
Figure 4 could be more informative and take up less space in the form of a table.
Figure A3 lacks a title on the YYs axis (“AIC”).
The last page of table A2 is unformatted. It could be simpler to provide this data as an independent data file.
The biometric measures of flower corollas and birds bills are not provided.

Experimental design

The study system selected (flower-bird visitor in the Peruvian Andes) is appropriate to explore the main question of the manuscript, which is well defined and falls within the scope of the journal. The methods are well described and the research follows a high technical standard.

Validity of the findings

The experimental design is not perfect but lends sufficient support to the main results. The methods and results are clear and robust.

Additional comments

General comments
Phenology was convincingly shown to be the best predictor for the observed interaction frequencies (observed matrix), however, Abundance is the best single predictor for 4 out of the 6 network metrics shown in figures 2 and 3 (outperforming Phenology in panels A, C, E and F). I believe that this results should be highlighted in the abstract and discussion even if the current title can be retained.

Sampling effort is estimated to have captured approximately half of the interactions in these communities (74 observed interactions out of 150 estimated; Figure A2). While this might not be too problematic when testing some hypothesis, it can be important in evaluating the relative importance of biological and neutral constraints, as biologically constrained interactions might be less common (e.g. interactions between coevolved specialists) and therefore less likely to be sampled under sub-optimal sampling efforts. My impression is that poor sampling is likely to over-emphasize the role of neutral constraints (for example for your results with connectance), but because you still picked a clear signal of biological constraints, more intensive sampling would likely confirm that signal. Please add a brief discussion of the potential effects of insufficient sampling effort on network metrics and specifically on the conclusions of this study.

Specific/minor comments

Given that the three experimental sites are so close together (600m – 1700m) and that the the same bird species are present on most sites due to high bird mobility, it might be easier and equally correct, to refer only to three plots, within a single experimental site. The geographic and biological proximity of the three plots fully justifies the option of merging the data into a single network.

L152. Sampling was not homogeneously spread around the year: some months have been sampled 3 times (July), while others were only sampled once (March). How can this affect the interpretation of plant phenology? This issue might not be too problematic but it might deserve a brief discussion.

L155. Improve sentence

L160. Were there any interactions observed on flowers not typically visited or suspected to be visited by birds? And if so, on what proportion? This information might be important to validate the option of focus the sampling effort only on ornithophilous plants. Birds can also consume nectar and pollen from unspecialized flowers and if these interactions are frequent they can potentially change the relative importance of biological and neutral constraints.

L188. I was a bit confused regarding the length of nets used per month. Presently the range presented is between 60m and 180m. If my math is right (probably isn’t), this correspond only to 1 net of 10 meters operated for 4 hour per site per month (on average). It was not clear why was the pollen load collected from birds’ beaks. Were pollen grains identified and incorporated into the dataset?

L385 and 395. Maybe replace “illegal” by “illegitimate”.

L386. ...through "the" corolla

L386. ... the "original" network

L406. I’m not familiarized with the expression “many fewer”; consider using “much less” instead.

L476. I didn’t understand this sentence. If these interactions were not observed, how can we infer that they are facilitated by the locally common flowerpiercers? I imagine that nectar robbing through holes in the corolla was observed for several short tongue hummingbirds, but when you say that these forbidden interactions did not occur, you mean that they didn’t result in actual pollination. Please clarify.

---

## Round 0.2 · Minor Revisions

Your ms was revised by one of the previous referees who still suggests a few minor revision.

I have also reviewed the ms myself, given that it is within my research lines, and I would like to add a few suggestions that you might consider taking into account when preparing the next version:

1-In the introduction, L 67-69, I’d suggest including a few more recent references supporting the statement that network structure influences stability and coevolutionary dynamics. There is indeed a hot debate on the relationship between some metrics like connectance, nestedness or modularity and network stability. See for instance, papers by Allesina et al. 2012, Suweis et al. 2013, Ruhr et al. 2015, James et al. 2015,

2-please change Vasquez for Vázquez throughout the ms. Also, H2 is usually written as H2’ in most papers using this network metric defined by Blüthgen et al. 2006.

3-Methods. L. 166-174. From 2011 through 2013 you use number of visits to plants as your link weight. In 2014, by contrast, you seem to score the number of flowers visited per unit time. You say that you combined such observations of birds from transects and focal plants to build the interaction network, but how do you do this, given that the link weights have been different? Also, it is not clear to me how you account for the number of different censuses you have done in each plant species, asyou say that multiple indiv plants and plant s pecies were simultaneously observed. Did you, each day ,census all flowering species in the area? In other words, how did you distribute the time of censuses across the different plant species? This is not clear at all in your methods.

4-L. 289-290. You may also want to look at the recent paper on the lability of forbidden links by González-Varo & Traveset TREE 2016 which refer to the existing intraspecific variation in morphological traits among individuals and the problem of inferring forbidden links from them.

5-L. 439. I am not sure whether you are aware of other more recent studies on flower-bird networks with a significant nested pattern, like Traveset et al. Nat Com. 2015.

6-L. 444-446. Since Bascompte et al. 2003, there has been a number of studies looking at the correlation between network size and nestedness, not always finding a positive value. You might thus want to check also more recent studies and reviews, or studies of global patterns (e.g. Sebastian-Gonzalez et al. GEB 2015).

I hope these suggestions and those of the reviewer are helpful to you when preparing the next version, which I look forward receiving it shortly.

Overall, I enjoyed reading your ms and I do think it is a valuable contribution to help assessing the relative important of neutral processes and biological constraints determining the structure of ecological networks.

Sincerely,
Anna Traveset

·

Basic reporting

There are a few sentences that, in my opinion, could be improved (see specific comments below), but overall the paper is clear, concise and informative.

Experimental design

There are some relevant methodological constraints in this paper, but I think that they are all well signaled in the discussion. In my opinion the authors successfully addressed the main concerns raised during the first revision, which further strengthened the manuscript.

Validity of the findings

Despite some methodological constraints, I believe that the main findings are robust and that the suggested approach is relevant and will stimulate further work in the area.

Additional comments

I still have a few minor suggestions that could help the authors improving the manuscript:

L61 All species in a network interact by definition with a subset of other species. But in a nested pattern specialist interact with subsets of the partners of the most generalist species. Please clarify.

L116 Consider “…at the base of the corolla”. To avoid repetition of “flowers” and also for greater clarity.

L150-151 Please revise this sentence.

L158-162 These section is still not reading very clearly, please revise. For example “… using transects to sites…”(?)

L403 Consider “entered the flower at the base”, instead of “at the nectar base”.

L424-5 That there are less interactions than those possible, particularly when only 55% of the species have been detected, it’s a truism. My concern in this sentence was specifically with the wording used: “many fewer” which I’m not sure it is the most correct, and not with the magnitude of the effect.

L432 This sentence starts with “For example”, but does not follow naturally from the previous sentence which was about the proportion of realized interactions while this one is about sampling effort. Please rephrase.

L452 But see also Costa et al. 2016, doi:10.1016/j.baae.2015.09.008 showing that, at least in some circumstances, WNODF can be more vulnerable to poor sampling than NODF.

L515 Correct “focal plant watches”.

Fig 1 The caption is incomplete.

Fig 2 and 3 There is no need to mention the metrics in the legend as they are visible in the figure.

---

## Round 0.3 · accepted · Accept

I thank you for revising again the ms, which I think is now acceptable for publication.

Good job and congratulations!

Yours sincerely,
Anna Traveset